# Two-Stream Convolutional Networks for Action Recognition in Videos

**Karen Simonyan**  **Andrew Zisserman**
Visual Geometry Group, University of Oxford
{karen,az}@robots.ox.ac.uk

## Abstract

We investigate architectures of discriminatively trained deep Convolutional Networks (ConvNets) for action recognition in video. The challenge is to capture the complementary information on appearance from still frames and motion between frames. We also aim to generalise the best performing hand-crafted features within a data-driven learning framework.

Our contribution is three-fold. First, we propose a two-stream ConvNet architecture which incorporates spatial and temporal networks. Second, we demonstrate that a ConvNet trained on multi-frame dense optical flow is able to achieve very good performance in spite of limited training data. Finally, we show that multi-task learning, applied to two different action classification datasets, can be used to increase the amount of training data and improve the performance on both. Our architecture is trained and evaluated on the standard video actions benchmarks of UCF-101 and HMDB-51, where it is competitive with the state of the art. It also exceeds by a large margin previous attempts to use deep nets for video classification.

## 1 Introduction

Recognition of human actions in videos is a challenging task which has received a significant amount of attention in the research community [11, 14, 17, 26]. Compared to still image classification, the temporal component of videos provides an additional (and important) clue for recognition, as a number of actions can be reliably recognised based on the motion information. Additionally, video provides natural data augmentation (jittering) for single image (video frame) classification.

In this work, we aim at extending deep Convolutional Networks (ConvNets) [19], a state-of-the-art still image representation [15], to action recognition in video data. This task has recently been addressed in [14] by using stacked video frames as input to the network, but the results were significantly worse than those of the best hand-crafted shallow representations [20, 26]. We investigate a different architecture based on two separate recognition streams (spatial and temporal), which are then combined by late fusion. The spatial stream performs action recognition from still video frames, whilst the temporal stream is trained to recognise action from motion in the form of dense optical flow. Both streams are implemented as ConvNets. Decoupling the spatial and temporal nets also allows us to exploit the availability of large amounts of annotated image data by pre-training the spatial net on the ImageNet challenge dataset [1]. Our proposed architecture is related to the two-streams hypothesis [9], according to which the human visual cortex contains two pathways: the ventral stream (which performs object recognition) and the dorsal stream (which recognises motion); though we do not investigate this connection any further here.

The rest of the paper is organised as follows. In Sect. 1.1 we review the related work on action recognition using both shallow and deep architectures. In Sect. 2 we introduce the two-stream architecture and specify the Spatial ConvNet. Sect. 3 introduces the Temporal ConvNet and in particular how it generalizes the previous architectures reviewed in Sect. 1.1. A mult-task learning framework is developed in Sect. 4 in order to allow effortless combination of training data over multiple datasets. Implementation details are given in Sect. 5, and the performance is evaluated in Sect. 6 and compared to the state of the art. Our experiments on two challenging datasets (UCF-101 [24] and HMDB-51 [16]) show that the two recognition streams are complementary, and our

deep architecture significantly outperforms that of [14] and is competitive with the state of the art shallow representations [20, 21, 26] in spite of being trained on relatively small datasets.

## 1.1 Related work

Video recognition research has been largely driven by the advances in image recognition methods, which were often adapted and extended to deal with video data. A large family of video action recognition methods is based on shallow high-dimensional encodings of local spatio-temporal features. For instance, the algorithm of [17] consists in detecting sparse spatio-temporal interest points, which are then described using local spatio-temporal features: Histogram of Oriented Gradients (HOG) [7] and Histogram of Optical Flow (HOF). The features are then encoded into the Bag Of Features (BoF) representation, which is pooled over several spatio-temporal grids (similarly to spatial pyramid pooling) and combined with an SVM classifier. In a later work [28], it was shown that dense sampling of local features outperforms sparse interest points.

Instead of computing local video features over spatio-temporal cuboids, state-of-the-art shallow video representations [20, 21, 26] make use of dense point trajectories. The approach, first introduced in [29], consists in adjusting local descriptor support regions, so that they follow dense trajectories, computed using optical flow. The best performance in the trajectory-based pipeline was achieved by the Motion Boundary Histogram (MBH) [8], which is a gradient-based feature, separately computed on the horizontal and vertical components of optical flow. A combination of several features was shown to further boost the accuracy. Recent improvements of trajectory-based hand-crafted representations include compensation of global (camera) motion [10, 16, 26], and the use of the Fisher vector encoding [22] (in [26]) or its deeper variant [23] (in [21]).

There has also been a number of attempts to develop a deep architecture for video recognition. In the majority of these works, the input to the network is a stack of consecutive video frames, so the model is expected to implicitly learn spatio-temporal motion-dependent features in the first layers, which can be a difficult task. In [11], an HMAX architecture for video recognition was proposed with pre-defined spatio-temporal filters in the first layer. Later, it was combined [16] with a spatial HMAX model, thus forming spatial (ventral-like) and temporal (dorsal-like) recognition streams. Unlike our work, however, the streams were implemented as hand-crafted and rather shallow (3-layer) HMAX models. In [4, 18, 25], a convolutional RBM and ISA were used for unsupervised learning of spatio-temporal features, which were then plugged into a discriminative model for action classification. Discriminative end-to-end learning of video ConvNets has been addressed in [12] and, more recently, in [14], who compared several ConvNet architectures for action recognition. Training was carried out on a very large Sports-1M dataset, comprising 1.1M YouTube videos of sports activities. Interestingly, [14] found that a network, operating on individual video frames, performs similarly to the networks, whose input is a stack of frames. This might indicate that the learnt spatio-temporal features do not capture the motion well. The learnt representation, fine-tuned on the UCF-101 dataset, turned out to be $20\%$ less accurate than hand-crafted state-of-the-art trajectory-based representation [20, 27].

Our temporal stream ConvNet operates on multiple-frame dense optical flow, which is typically computed in an energy minimisation framework by solving for a displacement field (typically at multiple image scales). We used a popular method of [2], which formulates the energy based on constancy assumptions for intensity and its gradient, as well as smoothness of the displacement field. Recently, [30] proposed an image patch matching scheme, which is reminiscent of deep ConvNets, but does not incorporate learning.

## 2 Two-stream architecture for video recognition

Video can naturally be decomposed into spatial and temporal components. The spatial part, in the form of individual frame appearance, carries information about scenes and objects depicted in the video. The temporal part, in the form of motion across the frames, conveys the movement of the observer (the camera) and the objects. We devise our video recognition architecture accordingly, dividing it into two streams, as shown in Fig. 1. Each stream is implemented using a deep ConvNet, softmax scores of which are combined by late fusion. We consider two fusion methods: averaging and training a multi-class linear SVM [6] on stacked $L_2$-normalised softmax scores as features.

**Spatial stream ConvNet** operates on individual video frames, effectively performing action recognition from still images. The static appearance by itself is a useful clue, since some actions are

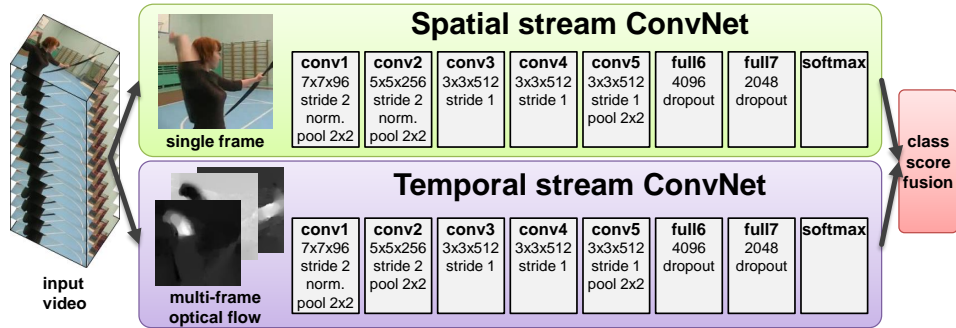

Figure 1: **Two-stream architecture for video classification.**

strongly associated with particular objects. In fact, as will be shown in Sect. 6, action classification from still frames (the spatial recognition stream) is fairly competitive on its own. Since a spatial ConvNet is essentially an image classification architecture, we can build upon the recent advances in large-scale image recognition methods [15], and pre-train the network on a large image classification dataset, such as the ImageNet challenge dataset. The details are presented in Sect. 5. Next, we describe the temporal stream ConvNet, which exploits motion and significantly improves accuracy.

## 3 Optical flow ConvNets

In this section, we describe a ConvNet model, which forms the temporal recognition stream of our architecture (Sect. 2). Unlike the ConvNet models, reviewed in Sect. 1.1, the input to our model is formed by stacking optical flow displacement fields between several consecutive frames. Such input explicitly describes the motion between video frames, which makes the recognition easier, as the network does not need to estimate motion implicitly. We consider several variations of the optical flow-based input, which we describe below.

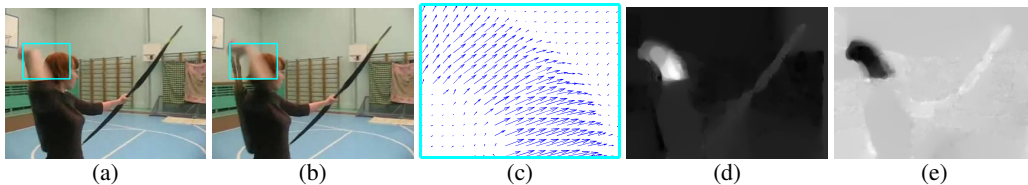

| (a) | (b) | (c) | (d) | (e) |

Figure 2: **Optical flow.** (a),(b): a pair of consecutive video frames with the area around a moving hand outlined with a cyan rectangle. (c): a close-up of dense optical flow in the outlined area; (d): horizontal component $d^x$ of the displacement vector field (higher intensity corresponds to positive values, lower intensity to negative values). (e): vertical component $d^y$. Note how (d) and (e) highlight the moving hand and bow. The input to a ConvNet contains multiple flows (Sect. 3.1).

### 3.1 ConvNet input configurations

**Optical flow stacking.** A dense optical flow can be seen as a set of displacement vector fields $\mathbf{d}_t$ between the pairs of consecutive frames $t$ and $t + 1$. By $\mathbf{d}_t(u, v)$ we denote the displacement vector at the point $(u, v)$ in frame $t$, which moves the point to the corresponding point in the following frame $t + 1$. The horizontal and vertical components of the vector field, $d_t^x$ and $d_t^y$, can be seen as image channels (shown in Fig. 2), well suited to recognition using a convolutional network. To represent the motion across a sequence of frames, we stack the flow channels $d_t^{x,y}$ of $L$ consecutive frames to form a total of $2L$ input channels. More formally, let $w$ and $h$ be the width and height of a video; a ConvNet input volume $I_\tau \in \mathbb{R}^{w \times h \times 2L}$ for an arbitrary frame $\tau$ is then constructed as follows:

$$I_\tau(u, v, 2k - 1) = d_{\tau+k-1}^x(u, v), \tag{1}$$
$$I_\tau(u, v, 2k) = d_{\tau+k-1}^y(u, v), \quad u = [1; w], v = [1; h], k = [1; L].$$

For an arbitrary point $(u, v)$, the channels $I_\tau(u, v, c), c = [1; 2L]$ encode the motion at that point over a sequence of $L$ frames (as illustrated in Fig. 3-left).

**Trajectory stacking.** An alternative motion representation, inspired by the trajectory-based descriptors [29], replaces the optical flow, sampled at the same locations across several frames, with

the flow, sampled along the motion trajectories. In this case, the input volume $I_\tau$, corresponding to a frame $\tau$, takes the following form:

$$I_\tau(u, v, 2k - 1) = d^x_{\tau+k-1}(\mathbf{p}_k), \qquad\qquad (2)$$
$$I_\tau(u, v, 2k) = d^y_{\tau+k-1}(\mathbf{p}_k), \quad u = [1; w], v = [1; h], k = [1; L].$$

where $\mathbf{p}_k$ is the $k$-th point along the trajectory, which starts at the location $(u, v)$ in the frame $\tau$ and is defined by the following recurrence relation:

$$\mathbf{p}_1 = (u, v); \qquad \mathbf{p}_k = \mathbf{p}_{k-1} + \mathbf{d}_{\tau+k-2}(\mathbf{p}_{k-1}), \; k > 1.$$

Compared to the input volume representation (1), where the channels $I_\tau(u, v, c)$ store the displacement vectors at the locations $(u, v)$, the input volume (2) stores the vectors sampled at the locations $\mathbf{p}_k$ along the trajectory (as illustrated in Fig. 3-right).

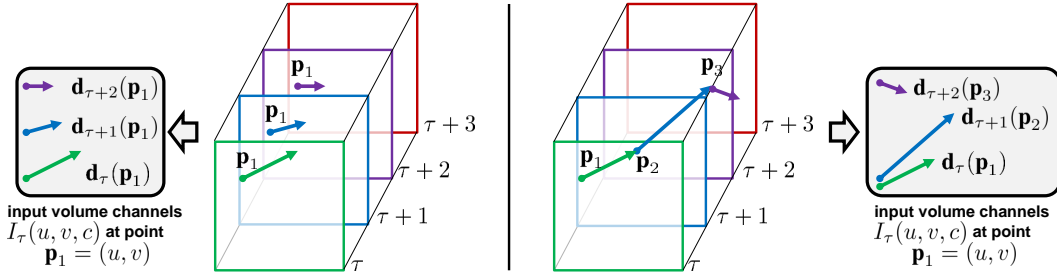

Figure 3: **ConvNet input derivation from the multi-frame optical flow.** *Left:* optical flow stacking (1) samples the displacement vectors $\mathbf{d}$ at the same location in multiple frames. *Right:* trajectory stacking (2) samples the vectors along the trajectory. The frames and the corresponding displacement vectors are shown with the same colour.

**Bi-directional optical flow.** Optical flow representations (1) and (2) deal with the forward optical flow, i.e. the displacement field $\mathbf{d}_t$ of the frame $t$ specifies the location of its pixels in the following frame $t + 1$. It is natural to consider an extension to a bi-directional optical flow, which can be obtained by computing an additional set of displacement fields in the opposite direction. We then construct an input volume $I_\tau$ by stacking $L/2$ forward flows between frames $\tau$ and $\tau + L/2$ and $L/2$ backward flows between frames $\tau - L/2$ and $\tau$. The input $I_\tau$ thus has the same number of channels $(2L)$ as before. The flows can be represented using either of the two methods (1) and (2).

**Mean flow subtraction.** It is generally beneficial to perform zero-centering of the network input, as it allows the model to better exploit the rectification non-linearities. In our case, the displacement vector field components can take on both positive and negative values, and are naturally centered in the sense that across a large variety of motions, the movement in one direction is as probable as the movement in the opposite one. However, given a pair of frames, the optical flow between them can be dominated by a particular displacement, e.g. caused by the camera movement. The importance of camera motion compensation has been previously highlighted in [10, 26], where a global motion component was estimated and subtracted from the dense flow. In our case, we consider a simpler approach: from each displacement field $\mathbf{d}$ we subtract its mean vector.

**Architecture.** Above we have described different ways of combining multiple optical flow displacement fields into a single volume $I_\tau \in \mathbb{R}^{w \times h \times 2L}$. Considering that a ConvNet requires a fixed-size input, we sample a $224 \times 224 \times 2L$ sub-volume from $I_\tau$ and pass it to the net as input. The hidden layers configuration remains largely the same as that used in the spatial net, and is illustrated in Fig. 1. Testing is similar to the spatial ConvNet, and is described in detail in Sect. 5.

### 3.2 Relation of the temporal ConvNet architecture to previous representations

In this section, we put our temporal ConvNet architecture in the context of prior art, drawing connections to the video representations, reviewed in Sect. 1.1. Methods based on feature encodings [17, 29] typically combine several spatio-temporal local features. Such features are computed from the optical flow and are thus generalised by our temporal ConvNet. Indeed, the HOF and MBH local descriptors are based on the histograms of orientations of optical flow or its gradient, which can be obtained from the displacement field input (1) using a single convolutional layer (containing

orientation-sensitive filters), followed by the rectification and pooling layers. The kinematic features of [10] (divergence, curl and shear) are also computed from the optical flow gradient, and, again, can be captured by our convolutional model. Finally, the trajectory feature [29] is computed by stacking the displacement vectors along the trajectory, which corresponds to the trajectory stacking (2). In the supplementary material we visualise the convolutional filters, learnt in the first layer of the temporal network. This provides further evidence that our representation generalises hand-crafted features.

As far as the deep networks are concerned, a two-stream video classification architecture of [16] contains two HMAX models which are hand-crafted and less deep than our discriminatively trained ConvNets, which can be seen as a learnable generalisation of HMAX. The convolutional models of [12, 14] do not decouple spatial and temporal recognition streams, and rely on the motion-sensitive convolutional filters, learnt from the data. In our case, motion is explicitly represented using the optical flow displacement field, computed based on the assumptions of constancy of the intensity and smoothness of the flow. Incorporating such assumptions into a ConvNet framework might be able to boost the performance of end-to-end ConvNet-based methods, and is an interesting direction for future research.

## 4    Multi-task learning

Unlike the spatial stream ConvNet, which can be pre-trained on a large still image classification dataset (such as ImageNet), the temporal ConvNet needs to be trained on video data – and the available datasets for video action classification are still rather small. In our experiments (Sect. 6), training is performed on the UCF-101 and HMDB-51 datasets, which have only: 9.5K and 3.7K videos respectively. To decrease over-fitting, one could consider combining the two datasets into one; this, however, is not straightforward due to the intersection between the sets of classes. One option (which we evaluate later) is to only add the images from the classes, which do not appear in the original dataset. This, however, requires manual search for such classes and limits the amount of additional training data.

A more principled way of combining several datasets is based on multi-task learning [5]. Its aim is to learn a (video) representation, which is applicable not only to the task in question (such as HMDB-51 classification), but also to other tasks (e.g. UCF-101 classification). Additional tasks act as a regulariser, and allow for the exploitation of additional training data. In our case, a ConvNet architecture is modified so that it has *two* softmax classification layers on top of the last fully-connected layer: one softmax layer computes HMDB-51 classification scores, the other one – the UCF-101 scores. Each of the layers is equipped with its own loss function, which operates only on the videos, coming from the respective dataset. The overall training loss is computed as the sum of the individual tasks' losses, and the network weight derivatives can be found by back-propagation.

## 5    Implementation details

**ConvNets configuration.** The layer configuration of our spatial and temporal ConvNets is schematically shown in Fig. 1. It corresponds to CNN-M-2048 architecture of [3] and is similar to the network of [31]. All hidden weight layers use the rectification (ReLU) activation function; max-pooling is performed over $3 \times 3$ spatial windows with stride 2; local response normalisation uses the same settings as [15]. The only difference between spatial and temporal ConvNet configurations is that we removed the second normalisation layer from the latter to reduce memory consumption.

**Training.** The training procedure can be seen as an adaptation of that of [15] to video frames, and is generally the same for both spatial and temporal nets. The network weights are learnt using the mini-batch stochastic gradient descent with momentum (set to 0.9). At each iteration, a mini-batch of 256 samples is constructed by sampling 256 training videos (uniformly across the classes), from each of which a single frame is randomly selected. In spatial net training, a $224 \times 224$ sub-image is randomly cropped from the selected frame; it then undergoes random horizontal flipping and RGB jittering. The videos are rescaled beforehand, so that the smallest side of the frame equals 256. We note that unlike [15], the sub-image is sampled from the whole frame, not just its $256 \times 256$ center. In the temporal net training, we compute an optical flow volume $I$ for the selected training frame as described in Sect. 3. From that volume, a fixed-size $224 \times 224 \times 2L$ input is randomly cropped and flipped. The learning rate is initially set to $10^{-2}$, and then decreased according to a fixed schedule, which is kept the same for all training sets. Namely, when training a ConvNet from scratch, the rate is changed to $10^{-3}$ after 50K iterations, then to $10^{-4}$ after 70K iterations, and training is stopped

after 80K iterations. In the fine-tuning scenario, the rate is changed to $10^{-3}$ after 14K iterations, and training stopped after 20K iterations.

**Testing.** At test time, given a video, we sample a fixed number of frames (25 in our experiments) with equal temporal spacing between them. From each of the frames we then obtain 10 ConvNet inputs [15] by cropping and flipping four corners and the center of the frame. The class scores for the whole video are then obtained by averaging the scores across the sampled frames and crops therein.

**Pre-training on ImageNet ILSVRC-2012.** When pre-training the spatial ConvNet, we use the same training and test data augmentation as described above (cropping, flipping, RGB jittering). This yields $13.5\%$ top-5 error on ILSVRC-2012 validation set, which compares favourably to $16.0\%$ reported in [31] for a similar network. We believe that the main reason for the improvement is sampling of ConvNet inputs from the whole image, rather than just its center.

**Multi-GPU training.** Our implementation is derived from the publicly available Caffe toolbox [13], but contains a number of significant modifications, including parallel training on multiple GPUs installed in a single system. We exploit the data parallelism, and split each SGD batch across several GPUs. Training a single temporal ConvNet takes 1 day on a system with 4 NVIDIA Titan cards, which constitutes a 3.2 times speed-up over single-GPU training.

**Optical flow** is computed using the off-the-shelf GPU implementation of [2] from the OpenCV toolbox. In spite of the fast computation time (0.06s for a pair of frames), it would still introduce a bottleneck if done on-the-fly, so we pre-computed the flow before training. To avoid storing the displacement fields as floats, the horizontal and vertical components of the flow were linearly rescaled to a $[0, 255]$ range and compressed using JPEG (after decompression, the flow is rescaled back to its original range). This reduced the flow size for the UCF-101 dataset from 1.5TB to 27GB.

## 6  Evaluation

**Datasets and evaluation protocol.** The evaluation is performed on UCF-101 [24] and HMDB-51 [16] action recognition benchmarks, which are among the largest available annotated video datasets[1]. UCF-101 contains 13K videos (180 frames/video on average), annotated into 101 action classes; HMDB-51 includes 6.8K videos of 51 actions. The evaluation protocol is the same for both datasets: the organisers provide three splits into training and test data, and the performance is measured by the mean classification accuracy across the splits. Each UCF-101 split contains 9.5K training videos; an HMDB-51 split contains 3.7K training videos. We begin by comparing different architectures on the first split of the UCF-101 dataset. For comparison with the state of the art, we follow the standard evaluation protocol and report the average accuracy over three splits on both UCF-101 and HMDB-51.

**Spatial ConvNets.** First, we measure the performance of the spatial stream ConvNet. Three scenarios are considered: (i) training from scratch on UCF-101, (ii) pre-training on ILSVRC-2012 followed by fine-tuning on UCF-101, (iii) keeping the pre-trained network fixed and only training the last (classification) layer. For each of the settings, we experiment with setting the dropout regularisation ratio to $0.5$ or to $0.9$. From the results, presented in Table 1a, it is clear that training the ConvNet solely on the UCF-101 dataset leads to over-fitting (even with high dropout), and is inferior to pre-training on a large ILSVRC-2012 dataset. Interestingly, fine-tuning the whole network gives only marginal improvement over training the last layer only. In the latter setting, higher dropout over-regularises learning and leads to worse accuracy. In the following experiments we opted for training the last layer on top of a pre-trained ConvNet.

**Temporal ConvNets.** Having evaluated spatial ConvNet variants, we now turn to the temporal ConvNet architectures, and assess the effect of the input configurations, described in Sect. 3.1. In particular, we measure the effect of: using multiple ($L = \{5, 10\}$) stacked optical flows; trajectory stacking; mean displacement subtraction; using the bi-directional optical flow. The architectures are trained on the UCF-101 dataset from scratch, so we used an aggressive dropout ratio of $0.9$ to help improve generalisation. The results are shown in Table 1b. First, we can conclude that stacking multiple ($L > 1$) displacement fields in the input is highly beneficial, as it provides the network with long-term motion information, which is more discriminative than the flow between a pair of frames

Table 1: **Individual ConvNets accuracy on UCF-101 (split 1).**

(a) **Spatial ConvNet.**

| Training setting | Dropout ratio | |
| --- | --- | --- |
| | 0.5 | 0.9 |
| From scratch | 42.5% | 52.3% |
| Pre-trained + fine-tuning | 70.8% | **72.8%** |
| Pre-trained + last layer | **72.7%** | 59.9% |

(b) **Temporal ConvNet.**

| Input configuration | Mean subtraction | |
| --- | --- | --- |
| | off | on |
| Single-frame optical flow ($L = 1$) | - | 73.9% |
| Optical flow stacking (1) ($L = 5$) | - | 80.4% |
| Optical flow stacking (1) ($L = 10$) | 79.9% | **81.0%** |
| Trajectory stacking (2)($L = 10$) | 79.6% | 80.2% |
| Optical flow stacking (1)($L = 10$), bi-dir. | - | **81.2%** |

($L = 1$ setting). Increasing the number of input flows from 5 to 10 leads to a smaller improvement, so we kept $L$ fixed to 10 in the following experiments. Second, we find that mean subtraction is helpful, as it reduces the effect of global motion between the frames. We use it in the following experiments as default. The difference between different stacking techniques is marginal; it turns out that optical flow stacking performs better than trajectory stacking, and using the bi-directional optical flow is only slightly better than a uni-directional forward flow. Finally, we note that temporal ConvNets significantly outperform the spatial ConvNets (Table 1a), which confirms the importance of motion information for action recognition.

We also implemented the "slow fusion" architecture of [14], which amounts to applying a ConvNet to a stack of RGB frames (11 frames in our case). When trained from scratch on UCF-101, it achieved 56.4% accuracy, which is better than a single-frame architecture trained from scratch (52.3%), but is still far off the network trained from scratch on optical flow. This shows that while multi-frame information is important, it is also important to present it to a ConvNet in an appropriate manner.

**Multi-task learning of temporal ConvNets.** Training temporal ConvNets on UCF-101 is challenging due to the small size of the training set. An even bigger challenge is to train the ConvNet on HMDB-51, where each training split is 2.6 times smaller than that of UCF-101. Here we evaluate different options for increasing the effective training set size of HMDB-51: (i) fine-tuning a temporal network pre-trained on UCF-101; (ii) adding 78 classes from UCF-101, which are manually selected so that there is no intersection between these classes and the native HMDB-51 classes; (iii) using the multi-task formulation (Sect. 4) to learn a video representation, shared between the UCF-101 and HMDB-51 classification tasks. The results are reported in Table 2. As expected, it is beneficial to

Table 2: **Temporal ConvNet accuracy on HMDB-51 (split 1 with additional training data).**

| Training setting | Accuracy |
| --- | --- |
| Training on HMDB-51 without additional data | 46.6% |
| Fine-tuning a ConvNet, pre-trained on UCF-101 | 49.0% |
| Training on HMDB-51 with classes added from UCF-101 | 52.8% |
| Multi-task learning on HMDB-51 and UCF-101 | **55.4%** |

utilise full (all splits combined) UCF-101 data for training (either explicitly by borrowing images, or implicitly by pre-training). Multi-task learning performs the best, as it allows the training procedure to exploit all available training data.

We have also experimented with multi-task learning on the UCF-101 dataset, by training a network to classify both the full HMDB-51 data (all splits combined) and the UCF-101 data (a single split). On the first split of UCF-101, the accuracy was measured to be 81.5%, which improves on 81.0% achieved using the same settings, but without the additional HMDB classification task (Table 1b).

**Two-stream ConvNets.** Here we evaluate the complete two-stream model, which combines the two recognition streams. One way of combining the networks would be to train a joint stack of fully-connected layers on top of full6 or full7 layers of the two nets. This, however, was not feasible in our case due to over-fitting. We therefore fused the softmax scores using either averaging or a linear SVM. From Table 3 we conclude that: (i) *temporal and spatial recognition streams are complementary, as their fusion significantly improves on both* (6% over temporal and 14% over spatial nets); (ii) SVM-based fusion of softmax scores outperforms fusion by averaging; (iii) using bi-directional flow is not beneficial in the case of ConvNet fusion; (iv) temporal ConvNet, trained using multi-task learning, performs the best both alone and when fused with a spatial net.

**Comparison with the state of the art.** We conclude the experimental evaluation with the comparison against the state of the art on three splits of UCF-101 and HMDB-51. For that we used a

Table 3: **Two-stream ConvNet accuracy on UCF-101 (split 1).**

| Spatial ConvNet | Temporal ConvNet | Fusion Method | Accuracy |
|---|---|---|---|
| Pre-trained + last layer | bi-directional | averaging | 85.6% |
| Pre-trained + last layer | uni-directional | averaging | 85.9% |
| Pre-trained + last layer | uni-directional, multi-task | averaging | 86.2% |
| Pre-trained + last layer | uni-directional, multi-task | SVM | **87.0%** |

spatial net, pre-trained on ILSVRC, with the last layer trained on UCF or HMDB. The temporal net was trained on UCF and HMDB using multi-task learning, and the input was computed using uni-directional optical flow stacking with mean subtraction. The softmax scores of the two nets were combined using averaging or SVM. As can be seen from Table 4, both our spatial and temporal nets alone outperform the deep architectures of [14, 16] by a large margin. The combination of the two nets further improves the results (in line with the single-split experiments above), and is comparable to the very recent state-of-the-art hand-crafted models [20, 21, 26].

Table 4: **Mean accuracy (over three splits) on UCF-101 and HMDB-51.**

| Method | UCF-101 | HMDB-51 |
|---|---|---|
| Improved dense trajectories (IDT) [26, 27] | 85.9% | 57.2% |
| IDT with higher-dimensional encodings [20] | **87.9%** | 61.1% |
| IDT with stacked Fisher encoding [21] (based on Deep Fisher Net [23]) | - | **66.8%** |
| Spatio-temporal HMAX network [11, 16] | - | 22.8% |
| "Slow fusion" spatio-temporal ConvNet [14] | 65.4% | - |
| Spatial stream ConvNet | 73.0% | 40.5% |
| Temporal stream ConvNet | 83.7% | 54.6% |
| Two-stream model (fusion by averaging) | 86.9% | 58.0% |
| Two-stream model (fusion by SVM) | **88.0%** | **59.4%** |

# 7  Conclusions and directions for improvement

We proposed a deep video classification model with competitive performance, which incorporates separate spatial and temporal recognition streams based on ConvNets. Currently it appears that training a temporal ConvNet on optical flow (as here) is significantly better than training on raw stacked frames [14]. The latter is probably too challenging, and might require architectural changes (for example, a combination with the deep matching approach of [30]). Despite using optical flow as input, our temporal model does not require significant hand-crafting, since the flow is computed using a method based on the generic assumptions of constancy and smoothness.

As we have shown, extra training data is beneficial for our temporal ConvNet, so we are planning to train it on large video datasets, such as the recently released collection of [14]. This, however, poses a significant challenge on its own due to the gigantic amount of training data (multiple TBs).

There still remain some essential ingredients of the state-of-the-art shallow representation [26], which are missed in our current architecture. The most prominent one is local feature pooling over spatio-temporal tubes, centered at the trajectories. Even though the input (2) captures the optical flow along the trajectories, the spatial pooling in our network does not take the trajectories into account. Another potential area of improvement is explicit handling of camera motion, which in our case is compensated by mean displacement subtraction.

## Acknowledgements

This work was supported by ERC grant VisRec no. 228180. We gratefully acknowledge the support of NVIDIA Corporation with the donation of the GPUs used for this research.

## Footnotes

[1]Very recently, [14] released the Sports-1M dataset of 1.1M automatically annotated YouTube sports videos. Processing the dataset of such scale is very challenging, and we plan to address it in future work.

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
