[Supplementary Material]

# Supplementary Material for the Paper "Two-Stream Convolutional Networks for Action Recognition in Videos"

**Karen Simonyan**                    **Andrew Zisserman**

Visual Geometry Group, University of Oxford
{karen,az}@robots.ox.ac.uk

## 1   Visualisation of learnt convolutional filters

In Fig. 1 we visualise the convolutional filters from the first layer of the temporal ConvNet, trained on the UCF-101 dataset. Each of the 96 filters has a spatial receptive field of $7 \times 7$ pixels, and spans 20 input channels, corresponding to the horizontal ($d^x$) and vertical ($d^y$) components of 10 stacked optical flow displacement fields **d** (the details can be found in Sect. 3 of the paper).

As can be seen, some filters compute spatial derivatives of the optical flow, capturing how motion changes with image location, which generalises derivative-based hand-crafted descriptors (e.g. MBH). Other filters compute temporal derivatives, capturing changes in motion over time.

Figure 1: **First-layer convolutional filters learnt on 10 stacked optical flows.** The visualisation is split into 96 columns and 20 rows: each column corresponds to a filter, each row – to an input channel.

## 2   Additional results on UCF-101

Here we report the average accuracy on the three splits of UCF-101, achieved using our temporal ConvNet, trained on UCF data only, i.e. without using the multi-task learning formulation. We also report the accuracy of the corresponding two-stream architectures. As can be seen, the performance is worse by $0.9\%$ than when additional HMDB training data is used, but it still remains competitive with the state of the art.

Table 1: **Mean accuracy (over three splits) on UCF-101.** The temporal stream model was trained on UCF-101 training data only, without additional HMDB data.

| Method | UCF-101 |
|---|---|
| Temporal stream ConvNet | 82.5% |
| Two-stream model (fusion by averaging) | 86.5% |
| Two-stream model (fusion by SVM) | **87.1%** |

# 3 Confusion matrix and per-class recall for UCF-101 classification

In Fig. 2 we show the confusion matrix for UCF-101 classification using our two-stream model, which achieves $87.0\%$ accuracy on the first dataset split (see the last row of Table 3 in the paper). We also visualise the corresponding per-class recall in Fig. 3.

The worst class recall corresponds to *Hammering* class, which is confused with *HeadMassage* and *BrushingTeeth* classes. We found that this is due to two reasons. First, the spatial ConvNet confuses *Hammering* with *HeadMassage*, which can be caused by the significant presence of human faces in both classes. Second, the temporal ConvNet confuses *Hammering* with *BrushingTeeth*, as both actions contain recurring motion patterns (hand moving up and down).

Figure 2: **Confusion matrix of a two-stream model on the first split of UCF-101.**

Figure 3: **Per-class recall of a two-stream model on the first split of UCF-101.**