[Reviews · NeurIPS 2014]

Submitted by Assigned_Reviewer_22

Summary:
This paper proposes a model for solving discriminative tasks with video inputs. The
model consists of two convolutional nets. The input to one net is an appearance
frame. The input to the second net is a stack of densely computed optical flow
features. Each pathway is trained separately to classify its input. The
prediction for a video is obtained by taking a (weighted) average of the
predictions made by each net.

The model is evaluated on two datasets. The results are impressive and match
(or come close to) the state-of-the-art methods which use intensively
hand-crafted features.

Strengths:
- This model is a simple application of convolutional nets that gets good results.
- Previous deep learning models have not tried to use optical flow (or other
hand-crafted features) for vision tasks, preferring to learn all features from
pixels directly. This paper shows that at least for action recognition, optical
flow fields are a useful input representation. This is an important contribution.

Weaknesses:
- The model does not take into account the overall sequence of
actions over the entire video but only models fixed length (L=10) adjacent frames.

Quality:
The experiments are well-designed. It would be more insightful if the authors also include some analysis of the error modes, for example, are their some classes or group of classes that the model is unable to classify well ? Is it possible to characterize the kind of videos on which the model does not work well ?

Clarity:
The paper is clearly written. The model is well explained.

Originality:
The application of a convolutional net to optical flow features is novel.

Significance:
This approach could have a significant impact on the research in using videos and motion for various vision problems.
Summary: The model is a simple application of convolutional nets to video data and gets very promising results. The use of optical flow features as inputs to a conv net is a novel contribution.

Submitted by Assigned_Reviewer_35

This paper presents a new neural network architecture for classifying videos of human actions. The model combines the predictions of two convolutional neural networks: one trained on single video frames and the other trained on short sequences of dense optical flow images. The model significantly outperforms other ConvNet-based approaches to action recognition and matches the current state-of-the-art on two standard video classification datasets.

While the paper essentially applies the standard ConvNet image classification pipeline to a new kind of data (dense optical flow frames) and combines this with a single frame ConvNet, the experiments are thorough and the results are impressive. It was good to see several different ways of training a ConvNet on optical flow compared with the results clearly showing the benefits of using optical flow as input.

I have only a few minor comments:
- It is not clear why the spatial network does so much better than the models from [13] given that the architectures are similar.
- It would be interesting to see a comparison of different sampling schemes for the frames fed into the spatial stream.
Summary: This is a good empirical paper that will be of interest to anyone working on video classification with ConvNets.

Submitted by Assigned_Reviewer_38

The paper proposes a few new methods for video classification with deep learning, namely two-stream formulation, and multi-task learning. The methodology, implementations, procedures were described in good detail. The paper is clear, well written, and presents a fair case for the proposed methods by showing good results.

These methods are novel. Although the paper adds improvements to the framework of [13], the two-stream framework is very interesting, because it shows that the optical flow component adds significant improvements. This is a result many researchers will be interested in. The paper is clearly presented, at the same time it seems to be split across multiple contributions as opposed to having a single theme. Finally, the results offer quite impressive improvements upon [13], and competitive with state-of-the-art.

Some suggestions to improve the paper:
1. It seems most of the significance is concentrated on the two-stream and incorporation of optical flow, the abstract and introduction could have a more focused theme.
2. Missing citations in deep learning:
Learning Hierarchical Spatio-temporal Features for Action Recognition with Independent Subspace Analysis. CVPR 2011.
Missing citations in computer vision:
Evaluation of local spatio-temporal features for action recognition. BMVC, 2010.
Summary: The paper seems to offer significant practical contributions, with some interesting ideas that complements the main ideas in deep learning. I would recommend to accept the paper at NIPS 2014, conditional upon that the authors provide the suggested improvements.
Author Feedback
Author rebuttal: We thank the reviewers for their comments.

R22: error modes

We briefly touch upon per-class performance in Sect. 3 of the Supp. material and will expand it in the final version.

R35: our spatial net performs better than the models of [13]

The reason for that could be the lower input resolution of [13] (170x170 instead of 224x224), as well as larger stride and smaller width of some convolutional layers in their case.

R38: make abstract and introduction more focused, add missing citations

We will improve the paper as suggested. Though note that the paper “Evaluation of local spatio-temporal features for action recognition” is actually already cited [24].